# Deconfounded Representation Similarity for Comparison of Neural Networks

**Tianyu Cui**
Department of Computer Science
Aalto University
tianyu.cui@aalto.fi

**Yogesh Kumar**
Department of Computer Science
Aalto University
yogesh.kumar@aalto.fi

**Pekka Marttinen**
Department of Computer Science
Aalto University
pekka.marttinen@aalto.fi

**Samuel Kaski**
Department of Computer Science
Aalto University and University of Manchester
samuel.kaski@aalto.fi

## Abstract

Similarity metrics such as representational similarity analysis (RSA) and centered kernel alignment (CKA) have been used to understand neural networks by comparing their layer-wise representations. However, these metrics are confounded by the population structure of data items in the input space, leading to inconsistent conclusions about the *functional* similarity between neural networks, such as spuriously high similarity of completely random neural networks and inconsistent domain relations in transfer learning. We introduce a simple and generally applicable fix to adjust for the confounder with covariate adjustment regression, which improves the ability of CKA and RSA to reveal functional similarity and also retains the intuitive invariance properties of the original similarity measures. We show that deconfounding the similarity metrics increases the resolution of detecting functionally similar neural networks across domains. Moreover, in real-world applications, deconfounding improves the consistency between CKA and domain similarity in transfer learning, and increases correlation between CKA and model out-of-distribution accuracy similarity.

## 1   Introduction

Deep neural networks (NNs) have achieved state-of-the-art performance on a wide range of machine learning tasks by automatically learning feature representations from data [1, 2, 3, 4, 5]. However, these networks do not offer interpretable predictions on most applications and are seen as "black boxes". It is thus crucial to understand the intricacies of neural networks before they are deployed on critical applications. Previous work has made progress in understanding how a single neural network makes decisions with axiomatic attribution methods [6, 7] and understanding how multiple neural networks relate to each other with representation similarity measures [8]. Several similarity measures between representations have been proposed with different principles, including linear regression [9], canonical correlation analysis (CCA; [10, 11]), statistical shape analysis [12], and functional behaviors on down-stream tasks [13, 14, 15]. Another main-stream approach is based on representational similarity analysis (RSA, [8, 16, 17, 18]) and centered kernel alignment (CKA, [19]), which compute the similarity between (dis)similarity matrices of two neural network representations on the same dataset.

RSA and CKA have been successfully applied to understand biological [20] and artificial NNs [21] by studying the similarity of *representations* of different NNs in a *single* data domain. However, we find

36th Conference on Neural Information Processing Systems (NeurIPS 2022).

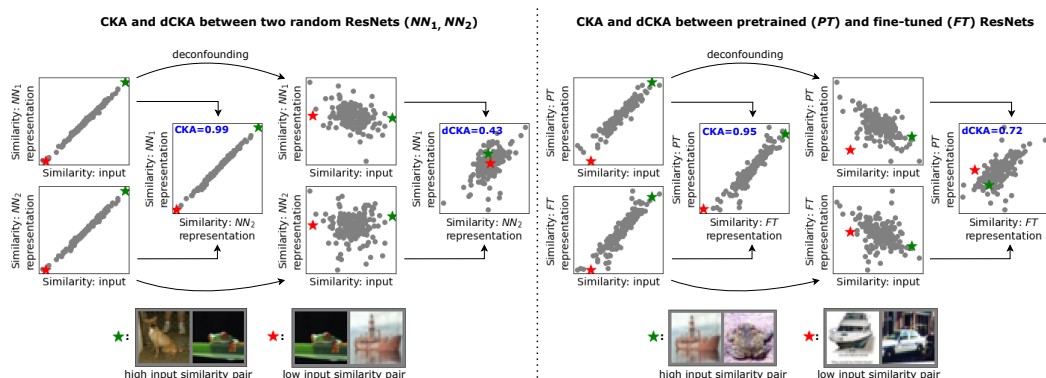

Figure 1: **Demonstration of the confounder in CKA.** CKA calculates the similarity between inter-example similarities for two representations, which are confounded by the inter-example similarities in the input space, such that input pairs with high (★) and low (★) input similarities also have high and low representation similarities on both random NNs (*Left*) and trained NNs (*Right*) representations. Moreover, the confounder leads to the counterintuitive conclusion that CKA on random NNs is higher than pretrained and finetuned NNs on similar domains (0.99 vs. 0.95). This is resolved by deconfounding (0.43 vs. 0.72).

that the inter-example (dis)similarity matrices in the representation space of different NNs are highly correlated with a shared factor (i.e., a confounder): the (dis)similarity structure of the data items in the input space, especially for shallow layers. This confounding issue limits the ability of CKA to reveal similarity of models on the *functional* level [15], and can result in a spuriously high CKA even between two random NNs. Moreover, this can lead to counter-intuitive conclusions when using CKA to compare models trained in *multiple* domains and/or to deduce *functional* similarities of NNs, such as in understanding transfer learning, multitask learning, and meta learning [22, 23, 24, 25, 26].

In this paper, we propose to adjust the representation similarity to improve its consistency w.r.t. the functional similarity by regressing out the confounder, the inter-example (dis)similarity matrix in the input space, from the (dis)similarity matrices of two representations, inspired by the covariate adjusted correlation analysis widely studied in biostatistics [27, 28]. This approach is simple and widely applicable on any similarity measure built on the CKA and RSA framework. Moreover, we study the invariance properties of the deconfounded representation similarity and demonstrate its benefits on public image and natural language datasets with various NN architectures.

Overall, our contributions are:

- We study the confounding effect of the input inter-example similarity on the representation similarity between two NNs, which limits CKA and RSA from revealing consistent *functional* similarities. We propose a simple and generally applicable deconfounding fix and discuss the invariance properties of the deconfounded similarities.

- We verify that deconfounded similarities can detect functionally similar NNs from random NNs, and small NN changes across domains where previous similarity measures fail.

- We show that deconfounded similarities are more consistent with domain similarities in transfer learning, on both image and language datasets, compared with existing methods.

- We demonstrate that deconfounded similarities on in-distribution datasets are more correlated with out-of-distribution accuracy than the corresponding original similarities [15].

## 2 Preliminaries

### 2.1 Notation and prior work

Let $X \in \mathbb{R}^{n \times p}$ denote the input dataset with $n$ datapoints and $p$ features, and $X_{f_1}^{m_1} \in \mathbb{R}^{n \times p_1}$ and $X_{f_2}^{m_2} \in \mathbb{R}^{n \times p_2}$ denote the $m_1$th and $m_2$th layer representations of two NNs of interest, $f_1(X)$ and $f_2(X)$, with $p_1$ and $p_2$ nodes respectively. We center and normalize the representation matrices by first removing the mean of each feature (i.e., each column), and then dividing by the Frobenius norm.

A standard approach for comparing representations of two NNs is to compare the similarity structures in each network representation. This can be done by first computing the similarity between every pair of examples in $X_{f_1}^{m_1}$ and $X_{f_2}^{m_2}$ with a similarity measure $k(\cdot, \cdot)$:

$$K_{f_1}^{m_1} = k(X_{f_1}^{m_1}, X_{f_1}^{m_1}), \; K_{f_2}^{m_2} = k(X_{f_2}^{m_2}, X_{f_2}^{m_2}). \tag{1}$$

Here $K_{f_1}^{m_1}, K_{f_2}^{m_2} \in \mathbb{R}^{n \times n}$ are called representational similarity matrices (RSMs)[1]. Second, another similarity measure $s(\cdot, \cdot)$ is applied to compare these two similarity structures, $K_{f_1}^{m_1}$ and $K_{f_2}^{m_2}$:

$$s_{f_1,f_2}^{m_1,m_2} = s(K_{f_1}^{m_1}, K_{f_2}^{m_2}). \tag{2}$$

This gives the similarity between the two representations.

The existing approaches vary by using different similarity measures for both levels of comparison. CKA [19] employs a kernel function for the first level similarity $k(\cdot, \cdot)$, and a Hilbert-Schmidt Independence Criterion (HSIC) estimator for the second, $s(\cdot, \cdot)$. On the other hand, RSA-based methods use Euclidean distance to measure the inter-example **dis**similarity structure, and apply Pearson's correlation [8] or Spearman's rank correlation [18] to quantify the similarity between two dissimilarity structures. Although other approaches, e.g., linear regression [29] and CCA [10], have been proposed to compare NN representations, we focus on CKA and RSA in this paper due to their wide usage in understanding the properties of NNs, such as transfer learning [22, 23, 30].

## 2.2 Illustration of the confounding in representation similarity

Given a dataset $X$ with an inter-example similarity matrix $K^0 = k(X, X)$ in the input space, making predictions with NNs can be seen as modifying $K^0$ layer-by-layer, such that similarities between data items with different labels decrease and those with the same label increase. In CKA and RSA, the similarity between two RSMs, $K_{f_1}^{m_1}$ and $K_{f_2}^{m_2}$, defines the similarity between two NNs, as shown in Eq.2. However, both $K_{f_1}^{m_1}$ and $K_{f_2}^{m_2}$ are affected by the same factor (i.e., a confounder): the similarity structure of input dataset $K^0$, which can cause spuriously high similarity. Intuitively, (dis)similar data points (green stars and red stars in Figure 1) in the input space are likely to be (dis)similar in the representation space of the first few layers, regardless of how the NN operates, and the representation similarity structure of different NNs would be similar even for random NNs that have totally different functional behaviors. Hence, CKA/RSA naturally depends on the specifics of the dataset, and for different datasets (e.g., from different domains) with different $K^0$, comparing CKA/RSA across the datasets may lead to inconsistent comparison results. This is undesirable especially when the goal of calculating the similarity measures for NNs is to quantify how similar the networks are in the functional level, which should not be significantly affected by the specifics of the dataset at hand.

We fix this by regressing out the input similarity $K^0$ from $K_{f_1}^m$ and $K_{f_2}^m$, i.e., deconfounding. After deconfounding, any similarity in the NN latent space between two data items would be induced by the NN and not because they were similar in the first place. This way, the deconfounded metric focuses more on comparing the functional form of the NNs and is less affected by the structure of the given dataset in the input space.

Figure 1 illustrates the spurious similarity using the CKA similarity measure as an example, and compares that with the deconfounded dCKA (defined in the next section) on the first-layer of ResNets [31] with 20 random samples from CIFAR-10 [32] test set. We consider two pairs of ResNets: 1. two random ResNets generated by adding different Gaussian noise, $\mathcal{N}(0, 1)$, to each parameter of the pretrained ResNet-18[2] on ImageNet [33]; 2. the pretrained (PT) ResNet-18 and a finetuned (FT) ResNet-18 on CIFAR-10. We notice that CKA on random NNs is almost 1, and counterintuitively it is even higher than the CKA between PT and FT ResNets on a similar domain (0.99 vs. 0.95), although we would expect the PT and FT networks to learn similar low-level features and hence be more similar than random networks. This happens because the similarities between samples in the input space confound their similarities in the representation space. After adjusting for the confounder with dCKA, the similarity between the two random ResNets is much smaller than the similarity of the PT and FT networks (0.43 vs. 0.72). A more detailed study is provided in Section 4.1.

---

[1]Note that $k(\cdot, \cdot)$ can also be dissimilarity measure, but we call $K_f^m$ a similarity matrix for simplicity.
[2]`https://pytorch.org/vision/stable/models.html`

# 3 Methods

In this section, we first propose a general fix on the NNs similarities (defined in Eq.1-2) by regressing out the confounder, i.e., the input similarity, in Section 3.1, and we give two examples of deconfounded similarities: deconfounded CKA and deconfounded RSA in Section 3.2. We then study the invariance properties of deconfounded similarities in Section 3.3.

## 3.1 Deconfounding representation similarity

We propose a simple approach to adjust the spurious similarity caused by the confounder by regressing out the input similarity structure from the representation similarity structure [34]. That is:

$$dK_{f_1}^{m_1} = K_{f_1}^{m_1} - \hat{\alpha}_{f_1}^{m_1} K^0, \ dK_{f_2}^{m_2} = K_{f_2}^{m_2} - \hat{\alpha}_{f_2}^{m_2} K^0, \tag{3}$$

where $K^0$ is the input similarity structure, and $\hat{\alpha}_{f_1}^{m_1}$ and $\hat{\alpha}_{f_2}^{m_2}$ are the regression coefficients that minimize the Frobenius norm of $dK_{f_1}^{m_1}$ and $dK_{f_2}^{m_2}$ respectively. Furthermore, the letter $d$ in front of a similarity matrix, e.g. as in $dK_{f_1}^{m_1}$, denotes the deconfounded version of $K_{f_1}^{m_1}$, and similarly the letter $d$ is applied throughout the text to denote all defounded quantities. To do the deconfounding, we assume that the input similarity structure $K^0$ has a linear and additive effect on $K_f^m$, i.e.,

$$\text{vec}(K_f^m) = \alpha_f^m \text{vec}(K^0) + \epsilon_f^m, \tag{4}$$

where $\text{vec}(\cdot)$ flattens a matrix to a vector. Noise $\epsilon_f^m$ is assumed to be independent from the confounder with $\hat{\epsilon}_f^m = \text{vec}(dK_f^m)$, and

$$\hat{\alpha}_f^m = (\text{vec}(K^0)^T \text{vec}(K^0))^{-1} \text{vec}(K^0)^T \text{vec}(K_f^m). \tag{5}$$

In Appendix A, we further verify that the linearity assumption in Eq.4 holds in general, i.e., adding additional nonlinear terms of $\text{vec}(K^0)$ to Eq.4 does not improve the fit, and the noise term $\epsilon_f^m$ is not auto-correlated (solution Eq.5 is not misspecified) under the the Durbin-Watson test [35].

After the deconfounded similarity structures are obtained with Eq.3, we use the same similarity measure to calculate the deconfounded representation similarity:

$$ds_{f_1,f_2}^{m_1,m_2} = s(dK_{f_1}^{m_1}, dK_{f_2}^{m_2}). \tag{6}$$

Note that $dK_f^m$ is not always positive semi-definite, even when $K_f^m$ is positive semi-definite (PSD). PSD is important for a similarity measure $s(\cdot, \cdot)$ that takes two kernel matrices as input, such as the CKA. We transform $dK_f^m$ into a positive semi-definite matrix by removing all the negative eigenvalues according to [36]. Specifically, we have the eigenvalue decomposition of $dK_f^m$, such that

$$dK_f^m = Q\Lambda Q^T = Q(\Lambda_+ - \Lambda_-)Q^T, \ \Lambda_\pm = \text{diag}\{\max(0, \pm\lambda_1), \ldots, \max(0, \pm\lambda_n)\}, \tag{7}$$

where $\lambda_i$ is the $i$th eigenvalue of $dK_f^m$. We approximate $dK_f^m$ with a PSD matrix $d\tilde{K}_f^m$:

$$dK_f^m \approx d\tilde{K}_f^m = \rho^2 Q\Lambda_+ Q^T; \ \ \rho = |\text{tr}(\Lambda)/\text{tr}(\Lambda_+)|. \tag{8}$$

## 3.2 Examples of deconfounded similarity indices

**Deconfounded CKA.** In CKA [19], the similarity structure in the feature space is represented with a valid kernel $l(\cdot, \cdot)$, i.e., $K_{f_1}^{m_1} = l(X_{f_1}^{m_1}, X_{f_1}^{m_1})$ and $K_{f_2}^{m_2} = l(X_{f_2}^{m_2}, X_{f_2}^{m_2})$, such as the linear or RBF kernel. Then an empirical estimator of HSIC [37] is used to align two kernels:

$$\text{HSIC}_{f_1,f_2}^{m_1,m_2} = \frac{1}{(n-1)^2} \text{tr}(K_{f_1}^{m_1} H K_{f_2}^{m_2} H), \tag{9}$$

where $H$ is the centering matrix. CKA is given by the normalized HSIC such that

$$\text{CKA}(K_{f_1}^{m_1}, K_{f_2}^{m_2}) = \text{HSIC}_{f_1,f_2}^{m_1,m_2} \Big/ \sqrt{\text{HSIC}_{f_1,f_1}^{m_1,m_1} \text{HSIC}_{f_2,f_2}^{m_2,m_2}}. \tag{10}$$

To deconfound the representation similarity matrices $K_{f_1}^{m_1}$ and $K_{f_2}^{m_2}$, we apply the same kernel to measure the inter-example similarity in the input space $K^0 = l(X, X)$, and adjust its confounding effect with Eq.3. However, matrices $dK_{f_1}^{m_1}$ and $dK_{f_2}^{m_2}$, obtained by regressing out one kernel matrix from another kernel, are no longer kernels, and they are not applicable for computing HSIC. Fortunately, with Eq.8, we can approximate the $dK_{f_1}^{m_1}$ and $dK_{f_2}^{m_2}$ with two valid kernels $d\tilde{K}_{f_1}^{m_1}$ and $d\tilde{K}_{f_2}^{m_2}$, which are then used to construct the deconfounded CKA (dCKA):

$$\text{dCKA}(K_{f_1}^{m_1}, K_{f_2}^{m_2}) = \text{CKA}(d\tilde{K}_{f_1}^{m_1}, d\tilde{K}_{f_2}^{m_2}). \tag{11}$$

We use linear kernels here because Kornblith et al. [19] report similar results with RBF kernels.

**Deconfounded RSA.** Different from CKA, the similarity structure in RSA [8] is measured by the pairwise Euclidean distance between examples in the feature space. Specifically, each element of $K_{f_1}^{m_1}$ and $K_{f_2}^{m_2}$ is obtained by $K_{f_1,ij}^{m_1} = \|\boldsymbol{x}_{f_1,i}^{m_1} - \boldsymbol{x}_{f_1,j}^{m_1}\|^2$ and $K_{f_2,ij}^{m_2} = \|\boldsymbol{x}_{f_2,i}^{m_2} - \boldsymbol{x}_{f_2,j}^{m_2}\|^2$, where $\boldsymbol{x}_{f_1,i}^{m_1}$ is the $m_1$-layer representation of the $i$th example in NN $f_1$. Thus, the input similarity structure $K^0$ is measured with the pairwise Euclidean distance in the input space. After $K^0$ is adjusted with Eq.3, we apply Spearman's $\rho$ correlation to measure the similarity between the upper triangular part of $dK_{f_1}^{m_1}$ and $dK_{f_2}^{m_2}$, i.e., $\text{triu}(dK_{f_1}^{m_1})$ and $\text{triu}(dK_{f_2}^{m_2})$, that is

$$\text{dRSA}(K_{f_1}^{m_1}, K_{f_2}^{m_2}) = \rho(\text{triu}(dK_{f_1}^{m_1}), \text{triu}(dK_{f_2}^{m_2})). \tag{12}$$

Note that rank correlation does not require two similarity matrices to be positive semi-definite. Therefore, we skip the steps of constructing the PSD approximation.

**Additional computational complexity.** The computational cost of the deconfounding (Eq.3) and PSD approximation (Eq.8) steps are $O(n^2)$ and $O(n^3)$ respectively. Therefore, dCKA has the same complexity as CKA: $O(n^3)$. For RSA, only the deconfounding step with $O(n^2)$ complexity is needed. In experiments, computing dCKA between two XLM-RoBERTa models [38] takes $0.37 \pm 0.11$s longer than CKA for each layer on 3000 random English sentences with a single 2080Ti GPU.

### 3.3 Theoretical properties

In this section, we study the invariance properties of the deconfounded representation similarity. For similarity measures that we studied, i.e., CKA and RSA, the corresponding deconfounded similarity indices have the same invariance properties, such as invariance to orthogonal transformation and isotropic scaling, as the original similarity measures.

**Proposition 3.1.** *Deconfounded CKA and deconfounded RSA are invariant to orthogonal transformation, if the (dis)similarity measure $k(\cdot, \cdot)$ that compare inter-examples are orthogonal invariant.*

**Proposition 3.2.** *Deconfounded CKA with a linear kernel and deconfounded RSA are invariant to isotropic scaling.*

Intuitively, as long as $k(\cdot, \cdot)$ is invariant to orthogonal transformation, e.g., linear kernels and Euclidean distance, the deconfounded representation similarity matrix $dK_f^m$ in Eq.3 is also invariant to orthogonal transformation, because it is defined in terms of the kernel $k$. Thus all operations on $dK_f^m$ are invariant to orthogonal transformation. Moreover, if one representation is scaled by a scalar, $dK_f^m$ and $d\tilde{K}_f^m$ will be scaled by the same scalar, whose effects will be finally eliminated in the normalization step in CKA (Eq.10) and the rank correlation step in RSA (Eq.12). We give proofs in Appendix B. A good functional similarity metric should be independent of the inter-example similarity in the input space (because that is dataset specific), but dependent on inter-example similarity in the representation space (because that is affected by the functional form of the NN). Above invariance properties ensure that the deconfounding does not sacrifice the desirable properties of CKA and RSA regarding inter-example similarities in the representation space, which are essential to understanding NNs in many cases [19].

## 4 Experiments

In this section, we design experiments to verify that deconfounding can improve the consistency of CKA and RSA with the functional similarity of NNs from various perspectives. Specifically,

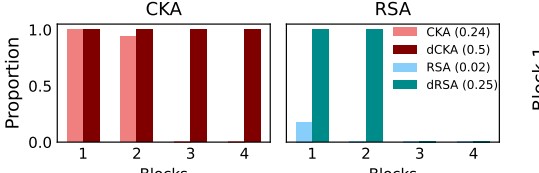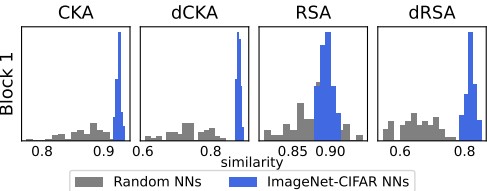

Figure 2: *Left*: **Proportion of ImageNet-CIFAR ResNets pairs identified from random ResNets for each block**. The proportions for the last four blocks are omitted as they are 0 for all metrics. The average proportions across layers are shown next to the method's name in the legend. We observe that deconfounding improves the identification of functionally similar NNs from random NNs. *Right*: **Histograms of similarities in the first block**. Compared with CKA and RSA, dCKA and dRSA increase separation of ImageNet-CIFAR pairs from the random network pairs, even when the proportion of identified network pairs with CKA and dCKA is the same.

we compare NNs layer-by-layer and then average the evaluation metrics about the consistency of layer-wise similarities to compare different similarity metrics [15]. First, in Section 4.1, we check if these representation similarity measures can separate functionally similar NN pairs from random NN pairs. In Section 4.2 we study how consistent the similarity measures are with small changes of NNs across different domains. We then extend a recent framework [15], which uses rank correlation to compare the representation similarity and the functional similarity of NNs, to transfer learning on multi-domain image and natural language datasets (Section 4.3), and to conduct challenging out-of-distribution generalization tests [15] (Section 4.4).

## 4.1 Ability of detecting functionally similar NN pairs from random NN pairs

**Setup:** We check if similarity measures can identify *functionally* similar NN representations from random NN representations. For each model block of ResNets (containing 2-3 convolutional layers), we generate two distributions of similarities: the null distribution $\mathcal{H}_0$ and the alternative distribution $\mathcal{H}_1$. The $\mathcal{H}_0$ contains similarities between 50 pairs of random ResNets on CIFAR-10 test set. We generate random NNs by randomly initializing the weight of an untrained ResNet-18 from $\mathcal{N}(0, 10)$ with different random seeds. We also consider permuting the weight matrix of the pretrained ResNet-18 to generate random NNs, which preserves parameter distributions, in Appendix D.1. Distribution $\mathcal{H}_1$ contains similarities between the pretrained ImageNet NN and each of the 50 ResNets trained on CIFAR-10 from scratch with different random initializations, on the same CIFAR-10 test as $\mathcal{H}_0$.

**Results:** We expect the similarities in $\mathcal{H}_1$ to be significantly larger than those in $\mathcal{H}_0$. Intuitively, models trained on ImageNet and CIFAR-10 are *functionally* similar in shallow blocks because their domains are similar; hence similar low-level features are expected [39], while sufficiently randomized untrained NNs are functionally different. We compute the proportion (shown in Figure 2 *Left*) of 50 ImageNet-CIFAR NN pairs in $\mathcal{H}_1$ whose similarity is significantly larger than $\mathcal{H}_0$, i.e., larger than the upper bound of its 95% CI. In Figure 2 *Left*, we observe that in shallow blocks the deconfounded similarity can detect a larger proportion of ImageNet-CIFAR pairs from random NN pairs than the original measures. For example, in the third and fourth block, dCKA can still detect all functionally similar pairs whereas the original CKA fails to detect any from random pairs. By averaging proportions of all blocks, deconfounding increases CKA from 0.24 to 0.5 and RSA from 0.02 to 0.25. In Figure 2 *Right*, we visualize the alternative distribution $\mathcal{H}_1$ and the null distributions $\mathcal{H}_0$ of each similarity measure as histograms in the first block. We observe that although CKA and dCKA can identify the same proportion of ImageNet-CIFAR pairs in block 1, the difference between $\mathcal{H}_1$ and $\mathcal{H}_0$ is more significant with the deconfounded similarities. Moreover, in deep layers, e.g., after block 4, no method can identify ImageNet-CIFAR pairs. We hypothesise that ImageNet and CIFAR-10 contain different classes of images, thus their high-level representations are significantly different. We discuss this more in Appendix D.

## 4.2 Consistency of NN functional similarities across domains

**Setup:** Ideally, a functional similarity between NNs would not depend on the domain in which the networks are applied. Here, we study this by constructing a set of 6 NNs, $\{f_i | i \in \{1, 2, 3, 4, 5, 6\}\}$,

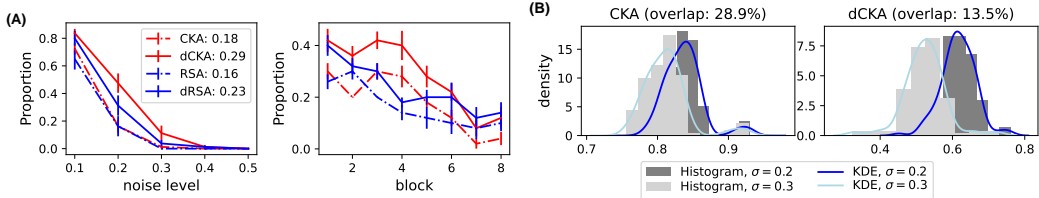

Figure 3: **(A): Proportion of identified similar NNs across 19 domains summarized for different noise levels (*Left*) and different NN blocks (*Right*).** We observe that the deconfounded similarity can identify more similar models compared with the corresponding original similarity. **(B): histograms and kernel density estimations (KDE) of CKA and dCKA across 19 domains on the first-block representations**. We observe that dCKA can separate $s(f_2, f^*)$ from $s(f_3, f^*)$ better than CKA with relative histogram overlap 13.5% and 28.9% respectively.

by adding independent Gaussian noise $i \times \mathcal{N}(0, 0.1)$ to each parameter of the pretrained ResNet-18 on ImageNet, $f^*$. Hence, the similarity $s(f_i, f^*)$ should be higher than $s(f_{i+1}, f^*)$ regardless of the input domain in which the the networks are applied to calculate representations. Thus, we calculate the similarity $s(f_i, f^*)$ for every $f_i$ on each domain of the corrupted CIFAR-10-C dataset [40] that contains 19 domains with different types of corruptions to the original CIFAR-10. We compute the average similarity $\mu_{f_i, f^*}$ across the 19 domains and its standard error $\sigma_{f_i, f^*}$. We say $f_i$ is significantly more similar to $f^*$ than to $f_{i+1}$ across domains, if

$$\mu_{f_i, f^*} - 1.96\sigma_{f_i, f^*} > \mu_{f_{i+1}, f^*} + 1.96\sigma_{f_{i+1}, f^*}. \tag{13}$$

We repeat the above experiments 20 times with different random seeds, i.e., generate 20 different sets of NNs, to measure the proportion of cases where $f_i$ is significantly more similar to $f^*$ than $f_{i+1}$ for each block, as well as the confidence interval of the proportion.

**Results:** In Figure 3(A) *Left*, we show the proportion of identified NNs averaged over all blocks for each noise level. We observe that the deconfounded similarity improves the proportion of identified NNs compared with the corresponding original similarity for all noise levels. The averaged proportion increases 59.7% for CKA (from 0.18 to 0.29) and 43.1% for RSA (from 0.16 to 0.23) after deconfounding. We also observe that deconfounding can improve CKA/RSA on different inputs from the *same* domain, but the improvement is relatively smaller (23% for CKA, from 0.65 to 0.8, and 7% for RSA, from 0.75 to 0.8), shown in Appendix E. Moreover, the proportion decreases as the noise level increases for all similarity measures, because for large noise level (large $i$), both $f_{i+1}$ and $f_i$ are far from $f^*$. In Figure 3(A) *Right*, we show the the proportion of consistently identified similarities for each block where the results are averaged over different noise levels. In general, we expect to identify fewer similar NNs with deeper layer representations, because the Gaussian noise is added to each parameter and deeper representations consequently accumulate more noise than shallow layers. We visualize the histogram of CKA and dCKA between $f_2$ and $f^*$ and between $f_3$ and $f^*$ on the first-block representations of inputs from 19 domains in Figure 3(B), where we can clearly observe that $f_2$ and $f_3$ are more separable in terms of dCKA than CKA. Moreover, the relative overlap areas between two histograms are 13.5% and 28.9% for dCKA and CKA respectively.

### 4.3 Transfer learning: domain similarity vs. the similarity of pretrained and finetuned NNs

**Motivation:** Ding et al. [15] argued that the similarity metric must be sensitive to changes that affect the *functionality* of the networks we compare, and we extend this to transfer learning under domain shift. Consider two models with the same initialization from a pretrained (PT) model which are finetuned (FT) on data from different domains. We then expect the similarity of the layer representations between the finetuned and pretrained models to be different for each target domain, and ideally it should be correlated with the similarity between the source and target domains.

**Setup:** To study this in detail for dCKA and CKA, we choose datasets from two modalities – image and text – that display such domain shift. For text, we use the Multi-lingual STS-B dataset [41] and choose English, Spanish, Polish, and Russian languages as the target domains. For images, we use the Real, Clipart, Sketch, and Quickdraw as target domains from the DomainNet dataset [42].

First, we finetune separate models for each domain from both modalities. For text, we initialize a PT XLM-RoBERTa model trained on 100 languages where the largest size of data is in English [38] and

Table 1: **Rank correlation with standard error (in parentheses) between the domain similarity and the CKA between pretrained and finetuned models**. We see that compared to CKA, the dCKA has higher correlation with domain similarity in terms of Spearman's $\rho$ and Kendall's $\tau$.

| Modality | CKA | | dCKA | |
| --- | --- | --- | --- | --- |
| | $\rho$ | $\tau$ | $\rho$ | $\tau$ |
| DomainNet | 0.675 (0.020) | 0.626 (0.018) | **0.751 (0.020)** | **0.718 (0.018)** |
| Multi STS-B | -0.231 (0.018) | -0.185 (0.016) | **0.717 (0.014)** | **0.641 (0.013)** |

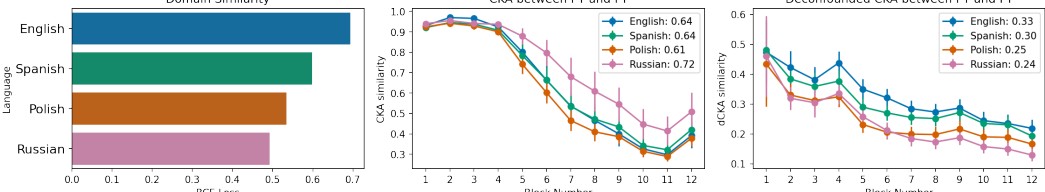

Figure 4: **dCKA adjusts for transfer learning under domain shift.** *Left* shows the ground truth domain similarity (between English and other languages) as measured by test binary cross entropy (BCE) loss of the cross-domain classifier. We plot the CKA (*Center*) and dCKA (*Right*) between the pretrained XLM-RoBERTa model and models finetuned for different languages on the STS-B task. The average similarities across layers are shown next to the languages in the legend. We observe that dCKA is better correlated with the domain similarity than CKA.

for images, we pretrain a ResNet-50 model on ImageNet [33]. We compute the layer-wise CKA and dCKA between each FT model and the corresponding PT model on the test set of the target domain [22]. We repeat the above procedure 10 times to construct error bars. Then, we quantify the similarity between two domains according to [43]. Specifically, we build a cross-domain classifier and create a dataset using an equal number of samples from each domain. Next, we train a weak discriminator to predict the appropriate true domain for each sample. For the discriminator, we use EfficientNet-B0 [44] for images and Distil-RoBERTa [45] for text. The discriminator is essentially a binary classifier and thus the test binary cross entropy (BCE) loss would be an indicator of the similarity between the two domains (higher meaning more similar). Finally, we measure the Spearman's $\rho$ and Kendall's $\tau$ correlation [15] between the domain similarity (measured by the BCE loss of cross-domain classifier) and the layer-wise CKA and dCKA between PT and FT models of each domain. We average the rank correlations over all layers.

**Results:** Table 1 summarizes the averaged rank correlations of CKA and dCKA for each modality. We can see that the dCKA is more correlated with the domain similarity as compared to CKA on both modalities. Figure 4 *Left* shows the results for the test BCE loss when the cross-domain classifier was trained to discriminate between the domains on STS-B: English-English, English-Spanish, English-Polish, and English-Russian, and we see that the similarity of two English domains is higher than the similarity of English and Russian domains, as expected. The layer-wise CKA and dCKA between the pretrained and finetuned XLM-RoBERTa model is shown Figure 4 (*Center* and *Right*), respectively. The models finetuned on highly dissimilar domains (e.g., English-Russian) are expected to have lower layer-wise similarity with the pretrained model. However, the CKA gives counter-intuitively high similarities between PT and FT for English-Russian, and negative rank correlations in Table 1. This is not because FT is close to PT, but because the Russian FT dataset is used to calculate the CKA, which causes the representations of PT and FT to be similar on that dataset. As expected, dCKA (Figure 4 *Right*) captures the domain similarity precisely by removing the input structure. We provide results on DomainNet in Appendix F.

### 4.4  Model generalization: in-distribution similarity vs. out-of-distribution accuracy

**Setup:** Here we use a challenge task used in previous work [15], which evaluates the sensitiveness of similarities to changes that affect the NNs generalizations on out-of-distribution (OOD) data, and we would expect similar NNs to have similar OOD accuracy. We follow the same setup as [15]: 1. We train 50 ResNet-18, $f_i$, with different random initialization on CIFAR-10; 2. Evaluate the OOD accuracy of each model on CIFAR-10-C [40], $\text{acc}(f_i)$, and select the most accurate ResNet as the

Table 2: **Rank correlation with standard error (in parentheses) between CKA and the prediction accuracy similarity of models**. We observe that dCKA improves correlations significantly on average, in terms of Spearman's $\rho$ and Kendall's $\tau$, with prediction accuracy on OOD test sets.

| Corruption level | CKA | | dCKA | |
|---|---|---|---|---|
| | $\rho$ | $\tau$ | $\rho$ | $\tau$ |
| 1 | 0.147 (0.004) | 0.103 (0.003) | 0.151 (0.004) | 0.105 (0.003) |
| 2 | 0.150 (0.004) | 0.106 (0.003) | 0.157 (0.004) | 0.110 (0.003) |
| 3 | 0.132 (0.004) | 0.094 (0.002) | **0.140 (0.003)** | **0.099 (0.003)** |
| 4 | 0.130 (0.003) | 0.091 (0.002) | **0.138 (0.003)** | **0.096 (0.003)** |
| 5 | 0.135 (0.003) | 0.094 (0.002) | 0.140 (0.004) | 0.098 (0.003) |
| Average | 0.139 (0.002) | 0.098 (0.001) | **0.145 (0.002)** | **0.102 (0.001)** |
| ID accuracy | 0.163 (0.020) | 0.116 (0.014) | 0.167 (0.020) | 0.118 (0.014) |

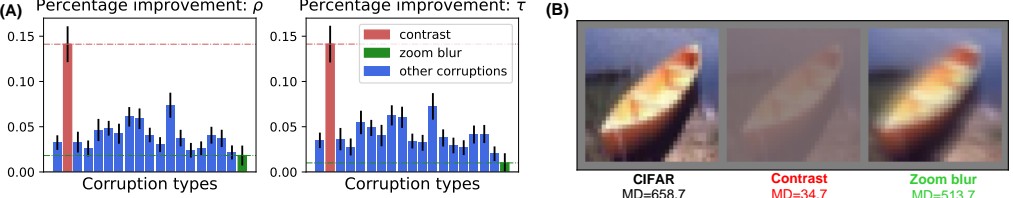

Figure 5: **dCKA improves CKA on each corruption type. (A):** percentage improvement of dCKA over CKA and corresponding standard error on each type of corruptions. **(B):** visualization of corruptions with largest improvement ('contrast', red) and smallest improvement ('zoom blur', green). We observe that 'contrast' is more different from the uncorrupted dataset compared with 'zoom blur', in terms of mean pair-wise distance (MD) of images.

reference model, $f^*$; 3. Compute the similarity between each $f_i$ and $f^*$, $s(f_i, f^*)$, of each block on CIFAR-10 test set (in-distribution similarity), and compute the accuracy difference on CIFAR-10-C test set $|\text{acc}(f_i) - \text{acc}(f^*)|$ (OOD accuracy similarity); 4. Measure the Kendall's $\tau$ and Spearman's $\rho$ between $1 - s(f_i, f^*)$ and $|\text{acc}(f_i) - \text{acc}(f^*)|$ for each block. A good similarity should have a high rank correlation, meaning that the similarity in the input space correlates with OOD accuracy.

**Results:** CIFAR-10-C dataset contains 19 different corruptions and 5 levels for each corruption. We average rank correlations over all blocks of ResNet-18 as [15] because the ranking between similarity measures were shown to be consistent across different layers/blocks. We report the averaged rank correlation over all types of corruptions in Table 2. We observe that dCKA is more correlated with OOD accuracy on all 5 levels of corruption, especially on levels 3-5, compared with CKA. Moreover, we also notice marginal improvements of dCKA in terms of in distribution accuracy (ID accuracy). Figure 5(A) shows the improvement of dCKA vs. CKA for each corruption averaged over 5 corruption levels, and we see that deconfounded CKA improves the most in the 'contrast' corruption (15%) and the least in the 'zoom blur' corruption (1%). We visualize examples of original in-distribution images together with the above two corruptions in Figure 5(B), and we observe that 'contrast' is very different from the in-distribution 'CIFAR' images whereas 'zoom blur' is more similar to the original images visually. Moreover, we compute the mean pairwise Euclidean distances (MD) between images for each type. We find the MD of the original CIFAR test set (MD= 658.7) is close to the 'zoom blur' (MD=513.7), and very far from the 'contrast' corruption (MD= 34.7). Hence, the benefit of dCKA vs. CKA appear greatest when the OOD domain is least similar to the original domain, which aligns with the expectation that dCKA is less domain specific due to the correction for the input domain population structure.

## 5 Discussion

We investigated the confounding effect of the input similarity structure on commonly used similarity measures between NNs representations. The confounder can lead to a high similarity even for completely random NNs and counter-intuitive conclusions when NNs trained in multiple domains are considered. We proposed a simple deconfounding algorithm by regressing out the input similarity from the representation similarity. The deconfounded similarity measures studied in this paper retain

the invariance properties of the original measures. Moreover, deconfounding significantly improves the consistency of similarities w.r.t. *functional* similarities of NNs, which is especially beneficial in understanding NNs when multiple domains are involved, such as studying the closeness between the finetuned and pretrained models in transfer learning (Figure 4). Although the deconfounded similarity does not improve models' performance directly, some insights can potentially inspire the future development of better machine learning models. For instance, in (low-resource) transfer learning, we could encourage the dCKA between the finetuned and pretrained models to be correlated with the known domain similarity during training, as we observed a high correlation between domain similarity and PT-FT similarity in Section 4.3.

We further clarify that the original CKA can provide a meaningful representation similarity in applications with a *single* domain, such as in understanding NN trained with different initializations [19] or with different architectures [21] on the same dataset, and in these cases dCKA is expected to yield similar conclusions with CKA (Appendix G). Moreover, in specific neuroscience applications [46], the input similarity structure can be a feature that should not be removed, and dCKA is not expected to be effective. However, in applications involving *multiple* domains, such as transfer learning, we showed that deconfounding yields more intuitive results, and we expect other applications with multiple domains or data sets, such as meta learning, to benefit from the insights in this paper.

There are still a few limitations and open questions. We assumed that the confounder is linearly separable from the representation similarity in Eq.4, and showed that adding higher-order polynomial terms cannot improve the model evidence in Appendix A. However, it is still possible that the input similarity structure is not entirely *additively* separable from representation similarity structures, especially for deeper layers, and this may explain the fact that deconfounding similarities are more beneficial for shallow layers (e.g., Figure 2 *Left* and Figure 3). One possible solution is regressing out the similarity structure in the previous layer instead of the input layer, which improves detecting similar networks from random networks for deep blocks (Appendix D.2). However, this discards information from all previous layers and eventually loses the ability of representing the similarity between functional behaviors. We consider this as an open question to motivate progress on developing more functionally consistent similarity measures.

## Acknowledgements

This work was supported by the Academy of Finland (Flagship programme: Finnish Center for Artificial Intelligence FCAI, and grants 292334, 315896, 336033), EU Horizon 2020 grants of European Network of AI Excellence Centres ELISE (951847) and INTERVENE (101016775), UKRI Turing AI World-Leading Researcher Fellowship (EP/W002973/1). We also acknowledge the computational resources provided by the Aalto Science-IT Project from Computer Science IT.

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
