# OpenReview forum: "Deconfounded Representation Similarity for Comparison of Neural Networks"
_NeurIPS.cc/2022/Conference — NeurIPS 2022 Accept_

### Official Review · Reviewer_4p6w · 2022-06-29

**Rating:** 6
**Confidence:** 3
**Soundness:** 4 excellent
**Presentation:** 4 excellent
**Contribution:** 3 good

**Summary:**

This paper aims at measuring the _functional_ similarity between two given neural networks.
The existing approaches naively compute some similarity measure of intermediate outputs of two neural networks, which may be largely entangled with input data, leading to spurious similarity.
The authors propose a method to deconfound the effect of input data by simply regressing out their effect.
The decounfounded similarity measures admit a good correlation with domain similarity and OOD accuracy, which are observed through experiments.

**Questions:**

### Questions

1. In Section 2.1 (prior work), the two-staged similarity measurement method is introduced (Eqs. (1) and (2)), but it may be not that evident why the first stage (Eq. (1)) is needed. A few more explanations would be preferred.
2. I am not sure why the invariance properties discussed in Section 3.3 are important. In the propositions, the proposed similarity measures are shown to be invariant against orthogonal transformations and isotropic scaling, which should not change the inter-example similarity. In my understanding, a good functional similarity should be independent of the inter-example similarity. If the aforementioned operations do not affect the inter-example similarity, the importance of the invariance would be questionable. Hence, I would like to see some clarification.

### Suggestions

1. In Eq. (2), do we need the full generality to introduce the similarity between different layers such as $m\_1$ and $m\_2$?
2. (minor) At l.88, "decounfounded" -> "deconfounded"
3. (minor) In Section 3.1, you may define $K^0$ properly right after Eq. (3).
4. (minor) In Figure 4 (right), having the averaged similarity in the legends (like Figures 2 and 3) should be better.

**Limitations:**

The authors adequately addressed the limitations in the discussion. The linearity assumption is one of the biggest limitations of this work, but the proposed method still works fairly in several experiments. This work may not be relevant to any negative societal impact.

**Strengths And Weaknesses:**

### Strengths

1. The spurious correlation between input data and functional similarity is formulated as a linear regression model. Although this simplicity may discard several more intricate structures, the deconfounded solution can be obtained by simply solving the normal equation. This formulation would be sufficient for the first step towards the deconfoundation.
2. The proposed method basically can be applied to any type of the existing representational similarity measures by applying the deconfoundation step. Through the experiments, we can observe that adding this simple fix is beneficial to improving the performance in most cases.

### Weaknesses

1. The motivation to measure the representational/functional similarity could be made a little bit clearer. For example, Section 4.3 provide the experiments to observe the correlation between the functional similarity and domain similarity, and Section 4.4 to see the correlation between the functional similarity and OOD accuracy. While these results may seem excellent, we would not immediately see what such a functional similarity brings us to improve transfer learning and OOD generalization.
2. The functional similarity is basically measured between two layers of given neural networks, not the entire neural networks. To measure the similarity between the entire networks, the authors simply average the layer-wise similarity through the experiments. Whereas this works in some situations (for example, Figure 2 tells us the averaged similarity is sufficient to distinguish random and fine-tuned ResNet-18), I have a concern when networks are deeper; most of the high-level features are similar between two networks, and hence the layer-wise averaged similarity would not be sufficient.

---

> ### Author Response · Authors · 2022-08-01
> **Response**
>
> Thank you for the valuable feedback!
>
> > Q1. The motivation to measure the representational/functional similarity [...] we would not immediately see what such a functional similarity brings us to improve transfer learning and OOD generalization.
>
> A: As Reviewer pUes mentioned, the main use case of the proposed method is in model interpretation as in previous works on similarity measures between NNs [1,2], and it can provide insights into how models work in meta or transfer learning problems. Moreover, we believe that some insights can potentially inspire the development of better ML models. For instance, in (low-resource) transfer learning, we could encourage the dCKA between the fine-tuned and pretrained models to be correlated with the known domain similarity during training, as we observed a high correlation between domain similarity and PT-FT similarity in Section 4.3.
>
> We will add the potential usages to the Discussion section (current page 9) in the camera-ready version.
>
> > Q2. To measure the similarity between the entire networks, the authors simply average the layer-wise similarity through the experiments [...] when networks are deeper; most of the high-level features are similar between two networks, and hence the layer-wise averaged similarity would not be sufficient.
>
> A: Actually we didn't average the layer-wise similarity to measure similarity between networks. Instead, by following the common practice, we compared NNs layer-by-layer  [2], and we averaged the evaluation metrics, such as the rank correlation, of layer-wise similarities to compare different similarity metrics [3]. This is mainly because the evaluation metrics have been shown to be consistent across layers in [3] and we observed similar behavior in our experiments too (e.g., Figure 4). Even when the evaluation metrics are not consistent across layers, it is still sensible to measure the average performance of each similarity metric across layers.
>
> When multiple domains/tasks are involved (the case that we focus more on), the learned high-level features from different domains can be very different (low similarities for deep layers in Figure 4).
>
> > Q3. In Section 2.1 (prior work), [...] why the first stage (Eq. (1)) is needed.
>
> A: By keeping the first stage, it is clearer to see that CKA is really a similarity of inter-example similarities in the representation space, and to calculate this, one first needs to calculate the representation similarities, which are calculated in Eq.1. Moreover, we can easily observe that the deconfounding step only affects this first stage. Thus, dCKA can be interpreted as a special case of CKA where the representation similarities are adjusted according to Section 3.1.
>
> > Q4. [...] why the invariance properties discussed in Section 3.3 are important. [...] A good functional similarity should be independent of the inter-example similarity. If the aforementioned operations do not affect the inter-example similarity, the importance of the invariance would be questionable.
>
> A: A good functional similarity metric should be independent of the inter-example similarity in the input space, i.e., the input similarity (because that is dataset specific), but dependent on inter-example similarity in the representation space (because that's affected by the functional form of the NN). The invariance properties ensure that the deconfounding does not sacrifice the desirable properties of CKA regarding inter-example similarities in the representation space, which are essential to understanding NNs in many cases (e.g., Q4 of Reviewer KLP6) [2].
>
> > Q5. In Eq. (2), do we need the full generality to introduce the similarity between different layers such as $m_1$ and $m_2$?
>
> A: We decided to keep this, because we consider the similarity between different layers in Appendix G.
>
> > Q6. (minor) At l.88, "decounfounded" -> "deconfounded"
>
> > Q7. (minor) In Section 3.1, you may define $K^{0}$ properly right after Eq. (3).
>
> > Q8. (minor) In Figure 4 (right), having the averaged similarity in the legends (like Figures 2 and 3) should be better.
>
> A: Thanks, corrected.
>
> **Reference**
>
> 1. Williams, Alex H., et al. Generalized shape metrics on neural representations. _NeurIPS_, 2021.
> 2. Kornblith, Simon, et al. Similarity of neural network representations revisited. _ICML_, 2019.
> 3. Ding, Frances, et al. Grounding Representation Similarity Through Statistical Testing. _NeurIPS_, 2021.

---

> > ### Comment · Reviewer_4p6w · 2022-08-04
> > **Thanks for the response**
> >
> > Many thanks for updating the manuscript and providing answers to my comments!
> >
> > > Q2
> >
> > Thanks for the clarification. Actually, I didn't understand that the authors "averaged the *evaluation metrics* of layer-wise similarities" when I first saw Figure 2, which is now very clear. I feel that this point (how to measure similarities between entire functions) can be stated clearly before going into the details of experiments (perhaps in Section 3) because the authors only provided a way to compute layer-wise similarities, while one of the final goals is to compare entire functions.
> >
> > > Q4
> >
> > Good point. I would prefer to see this explanation after the Propositions in the manuscript.

---

> > > ### Author Response · Authors · 2022-08-05
> > > **Thanks**
> > >
> > > Thank you for the good comments again!
> > >
> > > > Q2. Thanks for the clarification. Actually, I didn't understand that the authors "averaged the evaluation metrics of layer-wise similarities" when I first saw Figure 2, which is now very clear. I feel that this point (how to measure similarities between entire functions) can be stated clearly before going into the details of experiments (perhaps in Section 3) because the authors only provided a way to compute layer-wise similarities, while one of the final goals is to compare entire functions.
> > >
> > > A: Thanks for the valuable feedback. We will clarify the details of neural network comparison (i.e., layer-wise comparison) in the first paragraph of Section 4 on current page 5 in the camera-ready version.
> > >
> > > > Q4. Good point. I would prefer to see this explanation after the Propositions in the manuscript.
> > >
> > > A: Thanks. We will add the explanation to current Section 3.3 after the two propositions in the camera-ready version.

---

### Official Review · Reviewer_pUes · 2022-07-10

**Rating:** 7
**Confidence:** 4
**Soundness:** 4 excellent
**Presentation:** 3 good
**Contribution:** 3 good

**Summary:**

The paper investigates the confounding effect of similarity metrics used to compare neural networks on a layer-by-layer basis. The authors study the effect of this confounding effect. They argue that such measures as CKA and RSA are affected, thus, miscommunicating the functional similarity information by showing that the measures can show high similarity even for random neural networks. The paper proposes to 'deconfound' the metrics using a simple fix by regressing the input similarity structure from the representation similarity structure. Next, the authors conduct extensive experimentation to show the deconfounded measure performance under various settings.

**Questions:**

I don't have many questions as mostly the paper is well-written and provides all the necessary information. Just a few comments:

- In 3.3 you do not mention/link to the proofs of the proposition provided in the Appendix
- In figure 2 x-axis label is missing. Also, the colouring of Random NN-s is a bit difficult to see
- In 4.1 you perform a lot of experiments that include hypothesis testing and statistical significance. Have you considered multiple testing corrections?
- In Figure 5 (B) the colours for zoom blur and contrast are mislabeled.

**Limitations:**

No limitations detected

**Strengths And Weaknesses:**

Originality
The paper tackles an interesting problem, though, relatively narrow in scope, but still valuable for a meta or transfer learning domain. While the correction of the similarity metrics is not improving on models' performance, it can provide important insights into the inner workings of these models.

Quality
The paper is well-balanced and has all the necessary components: it introduces the problem and supports it with experiments, it solves the problem and provides necessary proofs without overburdening the paper. A multitude of experiments is introduced showing various aspects of the problem. The code and appendix are also provided. While some experiments are not straightforward and nested (having multiple steps in them on top of each other), I appreciate the complexity of the validation of the similarity metrics and how 'functional' similarity is difficult to define.

Clarity
It is a very well-written paper that is easy to read. It does not overcomplicate the idea, and the solution is quite elegant. I liked the experiments that provide interesting insights into neural networks' inner workings.

Significance
A relatively simple fix of the similarity metrics correction is not necessarily a major contribution that will lead to a groundbreaking impact in ML area, but it is a solid paper that helps to address a problem for future work on models' interpretation and lead to the better model training experience.

---

> ### Author Response · Authors · 2022-08-01
> **Response**
>
> Thank you for your good feedback!
>
> > Q1. In 3.3 you do not mention/link to the proofs of the proposition provided in the Appendix.
>
> A: Thanks, added!
>
> > Q2. In figure 2 x-axis label is missing. Also, the colouring of Random NN-s is a bit difficult to see.
>
> A: We now added the x-axis label and darkened the colouring of random NNs.
>
> > Q3. In 4.1 you perform a lot of experiments that include hypothesis testing and statistical significance. Have you considered multiple testing corrections?
>
> A: Good point; we now reduced the p-value threshold of hypothesis testing from $0.05$ to $\frac{0.05}{50}$ (50 hypotheses in
> total) according to the Bonferroni correction. The averaged proportions of identified NNs with CKA and RSA decreased from $0.24$ to $0.14$ and from $0.02$ to $0.01$ respectively, but dCKA ($0.5$) and dRSA ($0.25$) stayed unchanged, which highlights the benefit of deconfounding.
>
> > Q4. In Figure 5 (B) the colours for zoom blur and contrast are mislabeled.
>
> A: Thanks, corrected!

---

### Official Review · Reviewer_KLP6 · 2022-07-16

**Rating:** 7
**Confidence:** 4
**Soundness:** 3 good
**Presentation:** 3 good
**Contribution:** 3 good

**Summary:**

The paper presents an approach to improve CKA and RSA, which are the representational similarity metrics. Such metrics are used to compare representations from different layers of the same/different networks given the same list of objects.
Authors propose to "regress out"  inter-object similarity, the idea came from biostatistics.
Experiments with comparing networks trained from different seeds, fine-tuned networks, transfer learning show that the "deconfounded" CKA  (dCKA) is more sensitive than regular CKA and better aligns with the intuitive notion of functional similarity.
The paper is well written and easy to follow.

**Questions:**

1. eq. (4) what does vec(*) mean? I suppose it is matrix flattening? Maybe it makes sense to explain it explicitly.
2. The paper "Similarity of Neural Network Representations Revisited" describes an interesting experiment,
where the goal is to find the most similar layer between two networks trained from different seeds.
Ideally, the most similar layer should have the same number.
It is very interesting to evaluate your dCKA metrics for this problem.

3. line 132: "we can approximate dK_{f_1}^{m_1} and  dK_{f_1}^{m_1}" - you repeated dK_{f_1}^{m_1} two times.
4. in eq (3) you use an additional kernel to "regress out" the input similarity.  It should bw different from the original one in CKA.
Which kernels you used in the experiments?

**Limitations:**

No potential negative societal impact from this paper, in my opinion.

**Strengths And Weaknesses:**

Strengths:
* interesting idea regarding deconfounding.
* convincing experimental results from vision and text domains

Weakness:
* I wish authors explain the intuition behind deconfounding in more details.
* some experimental details are not presented - how do you choose kernels in dCKA?

---

> ### Author Response · Authors · 2022-08-01
> **Response**
>
> Thank you for the good comments!
>
> > Q1. The intuition behind deconfounding.
>
> A: We will fit the following paragraph into Section 2.2 (current page 3) in the camera-ready version.
>
> Suppose we have a set of data items with an inter-example similarity matrix $K^{0}$. Then, making predictions with NNs can be seen as modifying $K^{0}$ layer-by-layer, such that similarities between data items with different labels decrease and those with the same label increase. A comparison of two NNs, $f_1$ and $f_2$, with CKA/RSA is based on comparing $K_{f_1}^{m}$ and $K_{f_2}^{m}$, which are the inter-example similarity matrices on layer $m$ of $f_1$ and $f_2$. However, in practice both $K_{f_1}^{m}$ and $K_{f_2}^{m}$ are correlated to $K^{0}$: if two data items are very close in the input space, naturally, they are likely to be close in the latent space. Hence, CKA/RSA depends on the specifics of the dataset, and for different datasets (e.g., from different domains) with different $K^{0}$, comparing CKA/RSA across the datasets may lead to inconsistent comparison results. We fix this by regressing out the input similarity $K^{0}$ from $K_{f_1}^{m}$ and $K_{f_2}^{m}$. After deconfounding, if two data items are very similar in the NN latent space, it is because the NN made them so, and not because they were similar in the first place. In this way the deconfounded metric more directly focuses the comparison on the functional form of the NNs, and is less affected by the structure of the given dataset in the input space.
>
> > Q2. Some experimental details are not presented - how do you choose kernels in dCKA?
>
> A: We use the same kernel as CKA. More details below in the response to Q6.
>
> > Q3. eq. (4) what does vec(*) mean?
>
> A: Yes, it means matrix flattening. It is explained now.
>
> > Q4. The paper ``Similarity of Neural Network Representations Revisited" describes an interesting experiment, where the goal is to find the most similar layer between two networks trained from different seeds. Ideally, the most similar layer should have the same number. It is very interesting to evaluate your dCKA metrics for this problem.
>
> A: Actually we provided a similar experiment in Appendix G (with a short discussion in line 308 of the original paper). As expected, with dCKA, the most similar layers of models trained with different initializations usually have the same layer number  (17 out of 20 layers in ResNets trained on CIFAR-10). This is because CKA [1] and dCKA (Proposition 3.1) are both invariant to orthogonal transformation, and they are expected to behave similarly in this problem.
>
> > Q5. line 132: ``we can approximate $dK_{f_1}^{m_1}$ and $dK_{f_1}^{m_1}$" - you repeated $dK_{f_1}^{m_1}$ two times.
>
> A: Thanks, fixed!
>
> > Q6. In eq (3) you use an additional kernel to "regress out" the input similarity. Which kernels you used in the experiments to "regress out" the input similarity?
>
> A: The additional kernel $K^{0}$ is the same as $K_{f}^{m}$ (i.e. the original CKA kernel; mentioned in line 128 and line 139 of the original paper), though obviously calculated in the input space, whereas the CKA kernel is calculated in the representation space.
>
> **Reference**
>
> 1. Kornblith, Simon, et al. Similarity of neural network representations revisited. _ICML_, 2019.

---

> > ### Comment · Reviewer_KLP6 · 2022-08-08
> > **Response.**
> >
> > Thank you for clarifications.

---

### Author Response · Authors · 2022-08-01
**Response**

Thank you for the insightful and positive feedback. We are encouraged that you found our paper to be solid and well-written, and our problem interesting and valuable for interpreting models. We are glad that you found our solution to be simple and elegant, and our experiments extensive and convincing.  We believe we have been able to fix all concerns. We have provided a revised version for completeness, with changes marked in blue, and we will incorporate all feedback in the camera-ready version.

---

### Meta-Review · Area_Chair_HEka · 2022-08-28

**Recommendation:** Accept
**Confidence:** Certain

**Metareview:**

The paper makes the observation that neural network similarity indexes can be misleading when compared across domains with different examples. The paper presents a fix via covariate adjustment, which improves quality of similarity indexes across neural networks across domains. The approach is simple, and the reviewers unanimously agree that the paper is worthy of publication at NeurIPS.

**Award:**

No

---

### Decision · Program_Chairs · 2022-09-14

Accept